# Task-Relevant Covariance from Manifold Capacity Theory Improves Robustness in Deep Networks

**William Yang**[1,2*]    **Chi-Ning Chou**[1*]    **SueYeon Chung**[1,3]

[1]Flatiron Institute, Center for Computational Neuroscience, New York, NY, 10010
[2]University of Cambridge, Department of Engineering, Cambridge, UK, CB2 1PZ
[3]New York University, Center for Neural Science, New York, NY, 10003
[*]Equal contribution
{wyang, cchou, schung}@flatironinstitute.org

## Abstract

Analysis of high-dimensional representations in neuroscience and deep learning traditionally places equal importance on all points in a representation, potentially leading to significant information loss. Recent advances in *manifold capacity theory* offer a principled framework for identifying the computationally relevant points on neural manifolds. In this work, we introduce the concept of *task-relevant class covariance* to identify directions in representation-space supporting class discriminability. We demonstrate that scaling representations along these directions markedly improves simulated accuracy under distribution shift. Building on these insights, we propose AnchorBlocks, architectural modules that use task-relevant class covariance to align representations with a task-relevant eigenspace. By appending one AnchorBlock onto ResNet18, we achieve competitive performance in a standard domain adaptation benchmark (CIFAR-10C) against much larger robustness-promoting architectures. Our findings provide insight into neural population geometry and methods to interpret/build robust deep learning systems.

## 1   Introduction

In computational neuroscience and deep learning, geometric methods such as canonical correlation analysis (CCA) [1], representational similarity analysis (RSA) [2] and centered-kernel alignment (CKA) [3] have emerged as an effective approach for the analysis of high-dimensional neural representations. In such methods, point-cloud representations are often analyzed with a uniform prior, i.e., without explicitly privileging certain points over others. However, in their seminal work, Averbeck et al. [4] demonstrate how such uniformity assumptions on data points could lead to misestimation of information content relevant for downstream decoding. Overcoming this issue by optimally weighting data points is challenging in practice, which leaves us without a systematic framework for incorporating task-relevant information into the geometric analysis of neural representations.

*Manifold capacity theory* is a data-driven framework for quantifying task-relevant structure in neural representations through the lens of downstream readout by perceptrons [5, 6], and has found applications in both neuroscience [7, 8, 9, 6] and machine learning [10, 11, 12, 13, 14, 15, 16]. Manifold capacity generalizes the classic notion of storage capacity (e.g., Cover's theorem [17], Gardner's formula [18]) and quantifies the coding efficiency of neural representations in terms of the number of decodeable manifolds[1] stored per neuron. At the core of the theory is a mathematical result relating manifold capacity to a statistical distribution of *anchor points* (see Equation 2) which are the representative support vectors of each class manifold for a given decision boundary. Intuitively,

---

[1]Here, a neural manifold refers to the point cloud of representations produced by stimuli of the same class.

the anchor distribution can be understood as forming a *task-relevant geometry space* seen by an optimal linear classifier [6]. In particular, the task-relevant geometry produces class covariance structures that differ from those of the original point-cloud. As a toy example, Figure 1a depicts spherical class manifolds globally arranged in a ring structure[2]. For each class as well as globally, the principal directions of covariance are uninformative. However, if the goal of downstream neurons is to distinguish classes, then the task-relevant directions (red) are tilted, and would differ between class manifolds. This intuition is reflected by the covariance structure of the class anchor points (yellow).

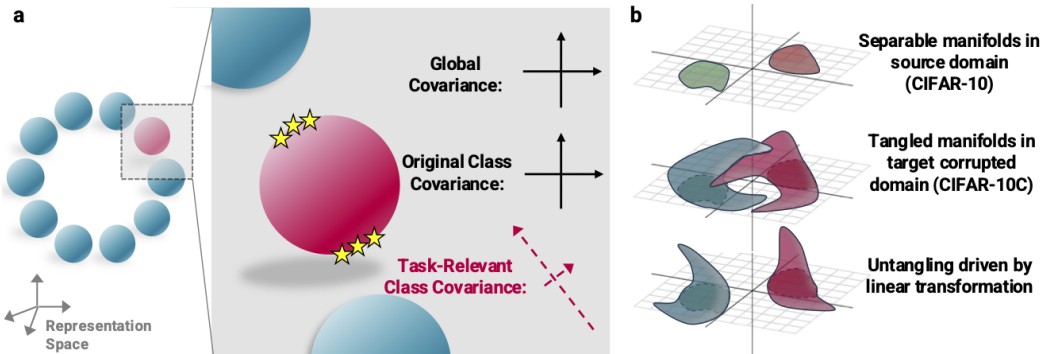

Figure 1: Schematic illustration. **a**, Anchor points (yellow) forming the task-relevant class manifold produce covariances which meaningfully differ from the original point-cloud manifold. **b**, Illustration of manifold tangling due to distribution shift, and subsequent untangling via linear transformation.

In this work, we apply these geometric insights to deep learning to investigate whether task-relevant class covariance from manifold capacity theory helps identify directions in representation-space useful for attenuating irrelevant or amplifying discriminative features. We focus on the setting of distribution shift, in which DNNs are vulnerable to even minor non-stationarities in input data, such as corruptions. We choose this setting due to extensive work, empirical and theoretical, attributing this vulnerability to a DNN's under-emphasis of auxillary class-invariant features, or over-emphasis on easily-degraded low-level features (see [22, 23, 24, 25, 26, 27, 28]). Using the standard CIFAR-10C domain adaptation task [29], we demonstrate the usefulness of task-relevant class covariance to identify linear transformations which improve robustness to distribution shift (see Figure 1b).

Our contributions can be summarized as follows:

- In Section 2.2, we formally introduce the concept of *task-relevant class covariance*, based on manifold capacity theory, as a way to identify directions in representation-space which support discriminability between classes.

- In Section 3.1, we find that manually augmenting ResNet18 representations with task-relevant class covariance improves simulated accuracy in domain adaptation on CIFAR-10C. In contrast, augmenting with the original point-cloud covariance is detrimental.

- In Section 3.2, we introduce *AnchorBlocks*, which are architectural modules designed to scale DNN representations along the eigenspace of task-relevant class-covariance matrices. We show that one AnchorBlock fine-tuned on a ResNet18 backbone achieves competitive domain adaptation on CIFAR-10C against larger robustness-promoting ResNet variants.

## 2 Methods

### 2.1 Manifold capacity theory and anchor geometry

Given a dataset $\{(\mathbf{x}_i, y_i)\}$ where $\mathbf{x}_i$ is the $i$-th input data and $y_i$ is its label, which comes from a finite discrete set $\mathcal{Y}$. We denote $\Phi(\mathbf{x}_i) \in \mathbb{R}^N$ as the representation of $\mathbf{x}_i$ (e.g. the hidden activation of a DNN) and $N$ as the representation dimensionality. For each class $y \in \mathcal{Y}$, the corresponding manifold $\mathcal{M}_y = \mathsf{conv}(\{\Phi(\mathbf{x}_i) : y_i = y\})$ is defined as the convex hull of the representations of the stimuli

---

[2]Although a toy example, ring-structured representations have been empirically observed, e.g., [19, 20, 21]

with label $y$. From manifold capacity theory [5, 6], the manifold capacity of $\Phi(\cdot)$ is denoted as $\alpha$, which exhibits a closed-form formula depending only on the structure of the manifolds as follows:

$$\alpha^{-1} = \mathop{\mathbb{E}}_{y \sim \mathsf{Unif}(\mathcal{Y})} \mathop{\mathbb{E}}_{\mathbf{g} \sim \mathcal{N}(0, I_N)} \left[ \max_{\substack{\mathbf{s}_{y'} \in \mathcal{M}_{y'} \\ \lambda_{y'} \geq 0, \forall y' \in \mathcal{Y}}} \left( \frac{-\sum_{y'} \delta(y, y') \lambda_{y'} \mathbf{g}^\top \mathbf{s}_{y'}}{\| \sum_{y'} \delta(y, y') \lambda_{y'} \mathbf{s}_{y'} \|_2} \right)^2_+ \right] \tag{1}$$

where $\delta(y, y') = 1$ if $y = y'$ and $-1$ otherwise, $\mathsf{Unif}(\cdot)$ is the uniform distribution of a finite set, vector $\mathbf{g}$ is drawn from the multivariate Gaussian distribution $\mathcal{N}(0, I_N)$, and $\lambda$ are Lagrange multipliers enforcing constraints of the convex optimization problem:

$$\min_{V \in \mathbb{R}^N} \|V - \mathbf{g}\|_2^2 \; : \; \delta(y, y') V^\top \mathbf{h} \leq 0, \; \forall y', \; \forall \mathbf{h} \in \mathcal{M}_{y'}$$

The term inside Eq. (1) has several interpretations, including the average error made by a random classifier. See [6] for further discussion and details. The key relevant observation is that Eq. (1) suggests a joint distribution over the manifolds $\{\mathcal{M}_y\}_{y \in \mathcal{Y}}$: $\{\mathbf{s}_{y'}(y, \mathbf{g})\}_{y' \in \mathcal{Y}} \sim \mathcal{D}_{\mathsf{anchor}}$ where $y \sim \mathsf{Unif}(\mathcal{Y})$ and $\mathbf{g} \sim \mathcal{N}(0, I_N)$. As such, we have:

$$\alpha^{-1} = \mathop{\mathbb{E}}_{\{\mathbf{s}_{y'}\}_{y' \in \mathcal{Y}} \sim \mathcal{D}_{\mathsf{anchor}}} [f(\{\mathbf{s}_{y'}\}_{y' \in \mathcal{Y}})] \tag{2}$$

where $f(\cdot) : \otimes_{y \in \mathcal{Y}} \mathcal{M}_y \to \mathbb{R}_{\geq 0}$ is a simple function as defined in Eq. (1). This establishes an analytical connection between manifold capacity $\alpha$ and the distribution of anchor points $\mathcal{D}_{\mathsf{anchor}}$.

## 2.2 Task-relevant covariance

Consider a dataset of representation-label pairs $\{(\mathbf{h}_i, y_i)\}$ for a class $y \in \mathcal{Y}$. We define the *point-cloud class covariance* as $C_y := \mathbb{E}_{\mathbf{h}_y \sim \mathsf{Unif}(\mathcal{H}_y)}[(\mathbf{h}_y - \bar{\mathbf{h}}_y)(\mathbf{h}_y - \bar{\mathbf{h}}_y)^\top]$, where $\mathcal{H}_y = \{\mathbf{h}_i : y_i = y\}$ and $\bar{\mathbf{h}}_y = \mathbb{E}_{\mathbf{h}_y \sim \mathsf{Unif}(\mathcal{H}_y)}[\mathbf{h}_y]$ is the mean representation of class $y$. In contrast, by focusing instead on the anchor points $\mathbf{s}_y$ induced by Eq. (2), we define the *task-relevant class covariance* of class $y$ as:

$$C_y^{\mathsf{eff}} := \mathop{\mathbb{E}}_{\{\mathbf{s}_{y'}\}_{y' \in \mathcal{Y}} \sim \mathcal{D}_{\mathsf{anchor}}} [(\mathbf{s}_y - \bar{\mathbf{s}}_y)(\mathbf{s}_y - \bar{\mathbf{s}}_y)^\top]. \tag{3}$$

## 2.3 Experimental setup

**Model.** All DNN experiments use a publicly-available pretrained ResNet18 [30] obtained from the Huggingface model repository [31] trained on CIFAR-10 [32] (see Section A.1).

**Data.** CIFAR-10C is used to evaluate robustness, and extends CIFAR-10 by algorithmically applying 15 types of common corruptions at 5 severity levels (1-5), resulting in 75 different test sets. Corruptions belong to four broad categories: noise, blur, weather and digital transformations [29].

**Robustness Metrics.** Per convention in [33], for each corruption we benchmark the *Corruption Error* (CE), the classification error rate, and *Mean Corruption Error* (mCE) which averages error over all 75 corruptions. We also report *Clean Error*, which is the error rate on the CIFAR-10 test set.

## 3 Results

We demonstrate the application of task-relevant covariance in two settings. In Section 3.1, we show that scaling the representations of a class along the eigenspace of their task-relevant class covariance matrix causes all class examples to be correctly classified even under distribution shift. In Section 3.2, we incorporate this geometric insight into a trainable DNN architecture which achieves competitive robustness on CIFAR-10C.

## 3.1 Representation Augmentation Experiment

If anchor geometry captures task-relevant class-specific features, then one prediction is that scaling representations along the eigenspace of their task-relevant class covariance should improve performance. To test this, we perform a controlled representation manipulation on pretrained ResNet18.

**Implementation.** Let $H_\mu^i \in \mathbb{R}^N$ be the representation of the $i$-th sample of the $\mu$-th class on a test set, where $i \in [M]$, $\mu \in [P]$. We first run pretrained ResNet18 in inference mode on the CIFAR-10 training set and extract activations used to compute a task-relevant covariance matrix for each class (see Eq. 3). We then run an inference pass on CIFAR-10C and augment each $H_\mu^i$ with the $\mu$-th task-relevant class covariance matrix $C_\mu^{\text{eff}}$, which can be summarized as $\mathbf{H}_\mu^{\text{Aug}} := \mathbf{H}_\mu C_\mu^{\text{eff}}$ (see Algorithm 1). To test the goodness of this augmented representation, we pass it through the model's readout to calculate simulated accuracy (Figure 2). As an experimental control, we perform the same procedure using the $\mu$-th point-cloud class covariance matrix from the CIFAR-10 training set.

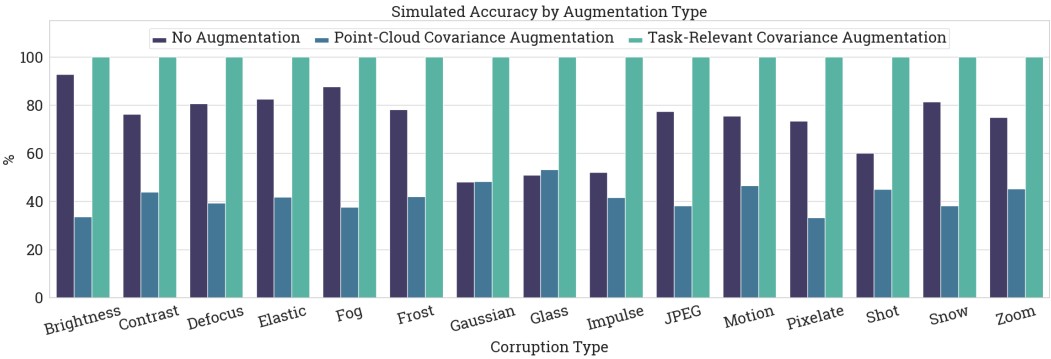

Figure 2: Simulated accuracy of representations augmented with different covariance matrices

**Augmentation with task-relevant class covariance results in perfect simulated accuracy.** Note that the manipulation used here is "label-aware", as the label determines which of the $P$ task-relevant class covariance matrices to use for a given test example. We use the term "simulated accuracy" to indicate that it is a measure used for analysis rather than to represent naturalistic classifier performance. We found representations achieve perfect simulated accuracy when scaled in the eigenspace of their task-relevant class covariance. Meanwhile, when representations are scaled by the point-cloud class covariance, simulated accuracy is worse than baseline. This is consistent with our intuition that task-relevant class covariance captures directions in feature-space supporting discriminability, whereas point-cloud class covariance merely captures directions of high within-class variation.

## 3.2 AnchorBlock

Although the procedure in Section 3.1 offers interesting insight relating task-relevant covariance to robustness, it is label-aware and thus would introduce a leakage if used in real-world settings. Here we introduce AnchorBlocks, architectural modules which allow models to learn to scale their representation by the correct task-relevant class covariance without label-assistance.

**Architectural summary.** We replace ResNet18's final layer with an AnchorBlock (see Figure 3a). For a $P$-class problem, the AnchorBlock has $P$ parallel *class covariance heads* with weights as task-relevant class covariance matrices, followed by a shared linear readout. The representation $\mathbf{h} \in \mathbb{R}^N$ is processed by all heads, yielding augmented vectors $\tilde{\mathbf{h}}_i$, $i \in [P]$. Passing these through the shared readout $W \in \mathbb{R}^{N \times P}$ gives $\mathbf{z}_i = W^\top \tilde{\mathbf{h}}_i$ for each $i = 1, \ldots, P$. By concatenating all $\mathbf{z}_i$, we form $Z = [\mathbf{z}_1, \ldots, \mathbf{z}_P] \in \mathbb{R}^{P \times P}$. The prediction is $\hat{\mathbf{y}} = \text{softmax}(\text{diag}(Z)) \in \mathbb{R}^P$. For full architectural details see Section A.3.

**Implementation.** Before training, we obtain the task-relevant class covariance matrices of all $P$ classes by running pre-trained ResNet18 on the CIFAR-10 training set in inference mode. The

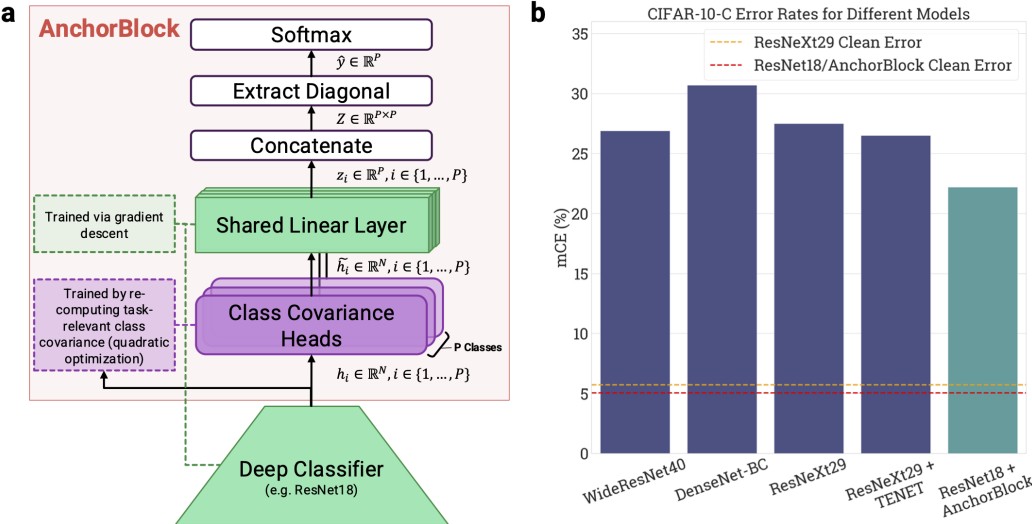

Figure 3: Overview of AnchorBlock. **a,** Schematic of AnchorBlock architecture. **b,** Mean Corruption Error (mCE) of robust ResNet variants. The mCE values for WideResNet, DenseNet and ResNeXt29 are reported in [33], while mCE for ResNeXt29 + TENET is reported in [27].

weights of each head are set to a class covariance matrix and kept fixed by disabling gradients. Then, we replace the readout of ResNet18 with one AnchorBlock, and fine-tune the combined model on the CIFAR-10 training set, keeping the objective function the same. This architecture is label-unaware since the representation is scaled by all $P$ task-relevant class covariance matrices in parallel.

**ResNet18+AnchorBlock achieves competitive performance compared to robust ResNet variants.** The task objective is to minimize error on CIFAR-10C using only CIFAR-10 training data. Following the convention of [33], we report the robustness of our classifier architecture using mCE (mean corruption error), which is simply the average classification error across all 75 CIFAR-10C test sets (see Figure 3b). The architectures we compare with are much larger, while DenseNet/RexNeXt implement enhanced feature aggregation - both of these properties purportedly promote robustness (see [29] for good discussion). In addition, our method also outperforms the combination of ResNeXt and TENET, an architectural modification designed to promote robustness and feature diversity [27].

## 4  Conclusion

In this work, we introduce the novel concept of *task-relevant covariance* derived from the statistics of anchor points in manifold capacity theory, and propose that they identify directions in representation space supporting class discriminability. In Section 3.1, we use a manual representation augmentation experiment to test our prediction about these discriminability properties, and find that simple linear transformations which scale representations along the eigenspace of task-relevant class covariance matrices markedly improve simulated accuracy on CIFAR-10C. In Section 3.2, we use these insights to create AnchorBlocks, trainable architectural modules which perform these linear transformations in a label-unaware manner. We find that fine-tuning one AnchorBlock on a ResNet18 backbone achieves competitive performance on CIFAR-10C compared to larger robustness-promoting architectures.

Our approach suggests a geometric interpretation of distribution shift as data non-stationarity that causes greater manifold tangling, in which case simple linear transformations may improve the robustness of representations via manifold untangling (see Figure 1b). In this geometric interpretation, task-relevant class covariance is a way to identify such linear transformations based on the geometry space seen by a downstream classifier. In a broader sense, our work emphasizes the importance of considering task-relevant structure when analyzing high-dimensional representations, and also highlights the differences of viewing representational statistics from the lens of an ideal observer versus a downstream perceptron. This perspective may have significant implications for the analysis of representations in both computational neuroscience and deep learning.

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

# A Appendix / supplemental material

## A.1 Experimental setup

**Data.** We use the CIFAR-10 and CIFAR-10C datasets. CIFAR-10C is used to evaluate model robustness, and extends CIFAR-10 by algorithmically applying 15 types of common corruptions at 5 levels of severity (1-5), resulting in 75 distinct test sets. Each type of corruption belongs to one of four categories: noise patterns, blurs, weather effects and digital transformations [29].

**Model.** Throughout this work, all our DNN experiments use ResNet [30]. Specifically, we use a publicly-available pretrained ResNet18 obtained from the Huggingface model repository [31] which was trained on CIFAR-10 [32] for 300 epochs using SGD optimizer implemented as `torch.optim.SGD(lr=0.1, momentum=0.9, weight_decay=0.0005, nesterov=True)` and a ReduceLROnPlateau scheduler.

**Fine-tuning.** When fine-tuning ResNet18 + AnchorBlock, we train the combined model on the CIFAR-10 training set for 50 epochs using an SGD optimizer implemented as `torch.optim.SGD(lr=0.01, momentum=0.5, weight_decay=0.01)`. Every 5 epochs, we update the weights of the class covariance heads by re-computing the effective class covariance matrices using the latest training representations.

**Computing task-relevant class covariance.** To compute task-relevant class covariance, we first run the model on the entire CIFAR-10 training set to obtain representations, which we extract from the last linear layer (dimensionality 512). These representations are in the form $\mathbf{H}_{\text{train}} \in \mathbb{R}^{P \times M \times N}$, where $P$ is the number of classes, $M$ is the number of points (or samples) per class, and $N$ is the dimensionality. Then we follow the standard procedure for computing anchor points described in [6]. This process involves Gaussian sampling of $K$ normal vectors and calculating the corresponding anchor points for each sample by solving a quadratic programming problem. In our experiments, we set `K = 200`. The process returns $K$ anchor points for each manifold, $\mathbf{S} \in \mathbb{R}^{P \times K \times N}$. Per Eq. (3), then for each class we compute the task-relevant class covariance matrix and stack them, which returns the task-relevant class covariance tensor $\mathbf{C}^{\text{eff}} \in \mathbb{R}^{P \times N \times N}$.

**Robustness metrics.** Per the convention in [33], for each corruption type we track the *Corruption Error* (CE), which is the classification error rate, $1 - \text{accuracy}$. To assess overall corruption robustness, we use *Mean Corruption Error*(mCE) which averages corruption error over all 75 corruption types. We also report *Clean Error*, which is the classification error rate on the original CIFAR-10 test set.

## A.2 Augmentation details

This section provides the detailed implementation on the augmentation used in Section 3.1, where we perform a controlled representation manipulation on the model's CIFAR-10C representations to test our intuitions about effective geometry. Let $H^i_\mu \in \mathbb{R}^N$ be the representation of the $i$-th sample of the $\mu$-th class. We run the pretrained model on CIFAR-10C and extract activations $\mathbf{H} \in \mathbb{R}^{P \times M \times N}$. Then, we scale each $H^i_\mu$ along the eigenspace of the $\mu$-th effective class covariance matrix (see Algorithm 1). For our control, we scale each $H^i_\mu$ along the eigenspace of the $\mu$-th point-cloud covariance matrix (see Algorithm 2)

---

**Algorithm 1:** Representation augmentation using task-relevant class covariance

**Data:** $\mathbf{H} \in \mathbb{R}^{P \times M \times N}, \mathbf{S} \in \mathbb{R}^{P \times K \times N}$
**Result:** $\mathbf{H}^{\text{Aug}} \in \mathbb{R}^{P \times M \times N}$ ;                    /* Augmented representation */
**for** $\mu \leftarrow 1$ **to** $P$ **do**

$\quad \mathbf{C}_\mu \leftarrow \text{Cov}(\mathbf{S}_\mu)$ ;  /* Covariance matrix of task-relevant class manifold */
$\quad \mathbf{H}^{\text{Aug}}_\mu \leftarrow \mathbf{H}_\mu \mathbf{C}_\mu$ ;            /* Scale $\mathbf{H}_\mu$ along eigenvectors of covariance */
$\quad \mathbf{H}^{\text{Aug}}_\mu \leftarrow \frac{\|\mathbf{H}_\mu\|_F}{\|\mathbf{H}^{\text{Aug}}_\mu\|_F} \mathbf{H}^{\text{Aug}}_\mu$ ;             /* Normalize to Frobenius norm of $\mathbf{H}_\mu$ */

---

**Algorithm 2:** Representation augmentation using point-cloud class covariance

**Data:** $\mathbf{H} \in \mathbb{R}^{P \times M_{\text{test}} \times N}$, $\mathbf{H}_{\mu}^{\text{train}} \in \mathbb{R}^{P \times M_{\text{train}} \times N}$

**Result:** $\mathbf{H}^{\text{Aug}} \in \mathbb{R}^{P \times M \times N}$ ;                    /* Augmented representation */

**for** $\mu \leftarrow 1$ **to** $P$ **do**

$\quad\mathbf{C}_{\mu} \leftarrow \text{Cov}(\mathbf{H}_{\mu}^{\text{train}})$ ;   /* Covariance matrix of point-cloud class manifold */

$\quad\mathbf{H}_{\mu}^{\text{Aug}} \leftarrow \mathbf{H}_{\mu}\mathbf{C}_{\mu}$ ;         /* Scale $\mathbf{H}_{\mu}$ along eigenvectors of covariance */

$\quad\mathbf{H}_{\mu}^{\text{Aug}} \leftarrow \frac{\|\mathbf{H}_{\mu}\|_F}{\|\mathbf{H}_{\mu}^{\text{Aug}}\|_F}\mathbf{H}_{\mu}^{\text{Aug}}$ ;              /* Normalize to Frobenius norm of $\mathbf{H}_{\mu}$ */

## A.3 AnchorBlock architecture details

In our setup, AnchorBlock replaces the final readout layer of a deep classifier. For a $P$-class classification problem, AnchorBlock consists of $P$ parallel "class covariance" heads followed by a shared linear readout layer. Each head is represented by a function $f_i : \mathbb{R}^N \to \mathbb{R}^N$ for $i = [P]$. In the forward pass, the representation vector $h \in \mathbb{R}^N$ is simultaneously processed by all heads. The resulting augmented representation vectors $\tilde{h}_1, \tilde{h}_2, ..., \tilde{h}_P$ are simultaneously passed through the shared readout layer parameterized by $W \in \mathbb{R}^{N \times P}$ to each produce vectors: $z_i = W^T \tilde{h}_i, z_i \in \mathbb{R}^P$. The $P$ vectors are then concatenated to form matrix $Z = [z_1, z_2, ..., z_P], Z \in \mathbb{R}^{P \times P}$. Finally, the diagonal elements of $Z$ are extracted to form the final prediction vector $\hat{y} = \text{softmax}(\text{diag}(Z)) \in \mathbb{R}^P$.

## A.4 Additional AnchorBlock results

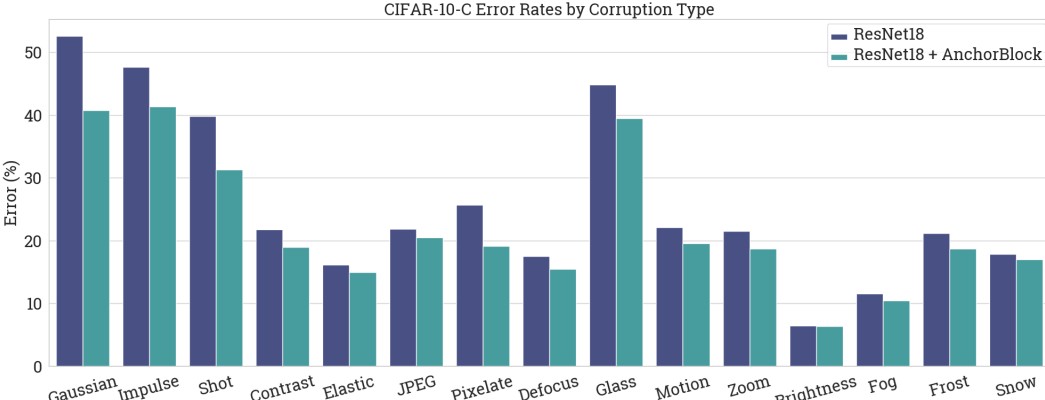

Figure 4: Comparison of mean corruption error between ResNet18+AnchorBlock and ResNet18.

