# OpenReview forum: "Task-Relevant Covariance from Manifold Capacity Theory Improves Robustness in Deep Networks"
_NeurIPS.cc/2024/Workshop/UniReps — UniReps_

### Official Review · Reviewer_fXby · 2024-10-05
**Review of Submission50**

**Rating:** 7
**Confidence:** 3

**Review:**

This paper proposes to exploit insight coming from manifold capacity theory to identify directions of maximal discriminability in representation space. Experimental results show that scaling representations along such directions produces promising results in classification under distribution shifts.

Strengths:

- The proposed method is interesting and well grounded in theory.
- Experimental results on domain adaptation, although preliminary, are very promising and already competitive with some previous solutions.
- Writing is clear, and experimental sections are well detailed.

Weaknesses:

- Results are reported in terms of mCE against other ResNet variants, showing that ResNet18 with AnchorBlock improves performance with respect to other models, in their standard version or trained with TENET. However, it would have helped to have at least a comparison with the current state of the art performances that can be obtained with models of this family. Moreover, it would be useful to assess the proposed method on additional datasets.
- The mathematical formulation in Eq. 1 is not very clear: e.g. the terms $\mathbf{g}$ and $\lambda$ that appear in the manifold capacity equation are never discussed in the text and their interpretation is not straightforward.
- (Minor) there is a typo in line 45 (auxillary -> auxiliary).

---

### Official Review · Reviewer_zsSR · 2024-10-06
**Review of submission #50**

**Rating:** 7
**Confidence:** 3

**Review:**

The main idea of this paper: introduces task-relevant class covariance to improve robustness in deep networks under distribution shift.
An important highlight, which I found unique was the idea they proposed of the AnchorBlock acheiving competitive performance to many benchmarks.

Strengths:
- Clear and well motivated writing -- task-relevant covariance derived from manifold capacity theory to enhance class discriminability.
- For initial stage results, they showed that ResNet18 + AnchorBlock outperforms larger architectures designed for robustness, such as ResNeXt + TENET.

Weaknesses:
- Would still like to see behaviour on a different dataset something more complex like CIFAR100.
- Incorporating some real-world distribution shifts (e.g., adversarial examples, real-world corruptions) would provide more practical insights into the effectiveness of task-relevant covariance for robustness. (this is more of question/suggestion to authors)

Question out of curiosity: is the AnchorBlock scalable to larger models and datasets? Did authors try any experiments regarding scalability?

---

### Official Review · Reviewer_udmU · 2024-10-06
**Review on TRC from MCT Improves Robustness in DN**

**Rating:** 6
**Confidence:** 3

**Review:**

This paper presents a novel method to enhance the robustness of deep neural networks when faced with distribution shifts by utilizing task-relevant class covariance based on manifold capacity theory.

Strengths
1. Application of Manifold Capacity Theory: The paper applies manifold capacity theory to identify task-relevant features, creating a connection between theoretical neuroscience concepts and deep learning.

2. Adaptability and Simplicity: The proposed AnchorBlocks represent a novel architectural feature that can be easily integrated into existing models, improving robustness without major architectural changes. Also it is easy to implement and do not require complex training processes, making the approach more feasible for practical applications.

3. Empirical Validation: The method demonstrates its effectiveness on a standard robustness benchmark (CIFAR-10C), delivering competitive performance against larger and more complex models focused on robustness.


Weaknesses
1. Comparison with State-of-the-Art Methods: The paper lacks comparisons with recent state-of-the-art robustness techniques, such as adversarial training or advanced data augmentation strategies, which could provide a better context for the method's performance.

2. Potential Computational Overhead: The introduction of multiple parallel class covariance heads could increase computational complexity, but the paper does not analyze or address these potential costs. An evaluation of computational efficiency during training and inference would be helpful.

3. Dependence on Label Availability for Training Covariance Matrices: While the AnchorBlock does not rely on labels during inference, it requires task-relevant class covariance matrices, which are computed using labels during training. This could limit the applicability of the method in scenarios with limited labeled data.



Overall, given the innovative approach, practical significance, and promising results, I recommend this paper for publication.

---

### Decision · Program_Chairs · 2024-10-10

**Decision:**

Accept

**Comment:**

In light of the positive reviewers' feedback and relevancy of the submission, we are pleased to accept this paper for presentation at UniReps 2024. We kindly ask the authors to incorporate the reviewers' suggestions and feedback in the final camera-ready version of the manuscript.